# Gold nanoparticle-enhanced radiotherapy: Dependence of the macroscopic dose enhancement on the microscopic localization of the nanoparticles within the tumor vasculature

**C. A. Díaz-Galindo, H. M. Garnica-Garza** [ORCID] *

Unidad Monterrey, Centro de Investigación y de Estudios Avanzados del IPN, Apodaca NL, México

* hgarnica@cinvestav.mx

## Abstract

In gold nanoparticle-enhanced radiotherapy, intravenously administered nanoparticles tend to accumulate in the tumor tissue by means of the so-called permeability and retention effect and upon irradiation with x-rays, the nanoparticles release a secondary electron field that increases the absorbed dose that would otherwise be obtained from the interaction of the x-rays with tissue alone. The concentration of the nanoparticles in the tumor, number of nanoparticles per unit of mass, which determines the total absorbed dose imparted, can be measured via magnetic resonance or computed tomography images, usually with a resolution of several millimeters. Using a tumor vasculature model with a resolution of 500 nm, we show that for a given concentration of nanoparticles, the dose enhancement that occurs upon irradiation with x-rays greatly depends on whether the nanoparticles are confined to the tumor vasculature or have already extravasated into the surrounding tumor tissue. We show that, compared to the reference irradiation with no nanoparticles present in the tumor model, irradiation with the nanoparticles confined to the tumor vasculature, either in the bloodstream or attached to the inner blood vessel walls, results in a two to three-fold increase in the absorbed dose to the whole tumor model, with respect to an irradiation when the nanoparticles have already extravasated into the tumor tissue. Therefore, it is not enough to measure the concentration of the nanoparticles in a tumor, but the location of the nanoparticles within each volume element of a tumor, be it inside the vasculature or the tumor tissue, needs to be determined as well if an accurate estimation of the resultant absorbed dose distribution, a key element in the success of a radiotherapy treatment, is to be made.

## 1. Introduction

The use of high-atomic number nanoparticles as radiation sensitizers has been well established in the past decade both in experimental [1, 2] as well as clinical studies [3–5]. Gold in

**Data Availability Statement:** All relevant data are accessible at DOI: https://doi.org/10.17026/PT/YSNGBI.

**Funding:** The work was funded by the National Council of Science and Technology of México (CONACyT) through grant F003-320355. One of the autthors, CAGD, received financial support from CONACyT to pursue Ph.D studies.

**Competing interests:** The authors have declared that no competing interests exist.

particular has been the focus of extensive research as it is a biologically compatible high-atomic number metal that can be synthesized as nanoparticles of various shapes, sizes and surface preparation [6]. Gold nanoparticles (GNPs) have a tendency to accumulate in tumors [7, 8], thus further sparing the healthy tissue around it from the toxic effect of the radiotherapeutic dose of ionizing radiation. For the case of irradiation with kilovoltage x-ray beams, upon absorption of an x-ray photon through the photoelectric effect, GNPs may emit Auger electrons and/or characteristic x-rays. Due to their high linear energy transfer (LET), Auger electrons have been shown to increase the radiobiological efficiency of the x-ray field resulting in higher tumoral cell killing for a given absorbed dose [9]. Therefore, two complementary effects are at play: the enhancement of the physical absorbed dose through the increase in the photoelectric absorption cross section and the enhancement of the cell killing efficiency of the secondary radiation field. In animal experiments, to which GNPs are intravenously administered, a third pathway to tumor damage is thought to occur at the vascular level, as the large concentration of GNPs in the bloodstream will result in substantial damage to the tumor vasculature after irradiation. This effect was recognized early on and, in fact, in the first report on the use of GNPs to enhance tumor dose in mice, the authors used Monte Carlo simulation to provide an initial estimate of the dose enhancement to the cells lining the tumor vasculature. They concluded that the enhancement was in the order of 550% [1]. As it has been established that absorbed doses in excess of 10 Gy imparted in a relatively short period of time are associated with severe vascular damage, thus resulting in indirect tumor cell death [10], considerable research has been devoted to study this third pathway to tumor cell killing. Lin *et al.* [11] compared, using Monte Carlo simulation, the dose enhancement in a simplified cylindrical model of a blood vessel that results from irradiation with protons and x-rays when a given concentration of GNPs is present either uniformly distributed in the blood or attached to the vessel wall, and found that for a prescribed dose of 2 Gy, local dose deposition events of at least 15 Gy to the vessel wall are possible. The dimensions of such hotspots were in the order of 10 to 70 nm in diameter. Ngwa *et al.* [12] calculated the dose enhancement to a simplified model of tumor endothelial cell nucleus that results when the irradiation is carried out with brachytherapy sources. They found nucleus dose enhancement factors in the range of 5 to 80 for GNP concentrations from 5 up to 40 mg/g. One obvious shortcoming of these studies is the simplified nature of the geometric models of both the blood vessel walls or cells and the distribution of the GNPs themselves. Rahman *et al.* [13] carried out experiments using bovine aortic endothelial cells to model endothelial tissue and measured cell survival as an indicator of the dose enhancement. They reported dose enhancement factors, using the 90% cell survival as the endpoint, in the order of 4 to 25 for concentrations ranging from 0.25 up to 1 mM when cells were irradiated with an 80 kVp x-ray beam. Using confocal microscopy, they also found that the GNPs, upon internalization by the endothelial cells, tend to form clusters in the cytoplasm. This clustering is consistent with reports using transmission electron microscopy [14] and x-ray microscopy [15] to image the GNP cellular uptake for different cell lines. The formation of GNP clusters has been shown to alter the resultant dose enhancement with respect to what would have been observed for the same GNP concentration without clustering [16, 17], and therefore it is important to consider this effect when modeling the irradiation of GNPs.

In this work, using a realistic tumor vasculature model, Monte Carlo simulation of the radiation transport and GNP distributions that consider clustering effects, it will be shown that the dose enhancement produced upon the interaction of x-ray with the GNPs depends not only on the total number of GNPs present in the irradiated volume but also on their spatial distribution within the bloodstream, the vessel walls and the surrounding tumor tissue. It will also be shown that, depending on the spatial distribution of the GNPs within the irradiated volume, the dose enhancement can be so high that it is possible to impart doses of radiation to the

blood vessel walls above the values at which vasculature failure has been reported to occur, using standard radiotherapy dose per fraction values.

## 2. Materials and methods

### a. Tumor vasculature model

The recursive rapidly exploring random tree algorithm (RRT) [18] was used as the starting point to model the characteristics exhibited by the capillary and microvasculature structures of solid tumors. RRT has been used before to model ductal tissue in breast models [19] or to model the structure of blood vessels in lung tumor models in mice [20]. The algorithm is designed to build a tree structure that randomly fills a given space, and it consists of three iterative steps: selection of an expansion vertex from an original point provided by the user, the expansion itself and some termination condition [21]. The algorithm was implemented in MatLab (Nantucket MA) with parameters chosen so as to model a realistic tumor microvasculature within a cube of 1mm in length divided into voxels with a resolution of 500 nm. The user provides two initial points on the surface of this cube, one is used as the entrance point corresponding to the tumor artery and the other is the venous return. Between each two random points, two concentric cylinders of the same length are placed, the outer cylinder with a radius 2 μm larger than the inner, in order to form the blood vessel walls. For every linear micrometer of added vasculature, the blood vessel radius is decreased by 500 nm. The termination condition was set as a function of the ratio of vasculature volume to total phantom volume, 3% according to the literature [22]. The space outside the vasculature tree is assumed to be tumor tissue, represented in this work by ICRU 44 soft tissue [23]. Other material explicitly modeled is the blood circulating in the tumor vasculature, with a composition taken also from ICRU Report 44. The tumor vasculature phantom was embedded in a soft tissue sphere with a radius large enough to achieve electronic equilibrium at all points inside the phantom, for all the energies modeled in this work. While the tumor vasculature model is small in size, a necessity dictated by its high resolution, the fact that it correctly reproduces the ratio of vasculature-to-tissue volume, that it is composed of biological materials, that it is embedded in a soft tissue sphere in order to achieve electronic equilibrium inside of it, and that it is, nevertheless, large enough to allow us to calculate absorbed dose at the dimensions typically used in radiotherapy treatment planning, makes this model suitable for the purpose of determining the effect that the presence of GNPs have on the absorbed dose imparted not only to the tumor tissue but also to the blood vessel walls.

### b. Monte Carlo simulations

The Monte Carlo code PENELOPE [24] with the auxiliary set of subroutines from the PenEasy suit [25] was used throughout this work. Following the recommendations of the code developers, in order to accurately model the transport of electron in micrometer sized dimensions, the condensed history algorithm was disabled by setting the transport parameters C1 and C2 to zero and, therefore, all the results presented in this work were obtained using step-by-step electron transport. Cutoff energies were set at 100 eV for both photons and electrons. The standard photon and electron interaction cross section PENELOPE libraries were used. In each simulation, at least $1\times10^{11}$ histories were run in order to obtain an average statistical uncertainty in the order of 2%. Depending on the energy spectrum, each simulation took between 600 and 900 hours, running on an AMD Ryzen 7 2700x processor at 3.7 GHz.

A GNP concentration of 10 mg-Au/g was assumed to be present in the vasculature phantom developed, a concentration representative of the range of values observed clinically [3]. As mentioned above, the GNPs tend to form clusters of irregular shape with a size in the order of

500 nm [13–15], and therefore it was assumed in this work that the GNPs form clusters of this size. This assumption also allows for the explicit simulation of each and every GNP and a gold density of 19.3 g/cm$^3$ was used to determine the number of GNPs necessary to achieve the 10 mg-Au/g concentration of gold in the vascular phantom. Three situations of interest were separately evaluated: GNPs circulating exclusively in the blood, GNPS attached to the internal blood vessel walls, and GNPs uniformly distributed in the tumor tissue. While it is likely that once GNPs extravasate from the vasculature and diffuse in the surrounding tissue a concentration gradient would form, this would only be the case if the blood vessels are rectilinear and fairly parallel to each other. For the inherently random tumor vasculature, we decided that the most representative approach would be to consider a uniform spatial distribution of the GNPs in the tumor tissue. Three photon spectra were used in our simulations, namely, a 6 MV x-ray beam from a flattening filter-free medical linear accelerator [26], a 220 kVp x-ray beam previously shown to be the optimal spectrum for a variety of treatment sites [27], and a monoenergetic 100 keV photon beam chosen to maximize the photoelectric effect probability.

Using patient images derived from Dynamic Contrast-Enhanced MRI studies (DCE-MRI), we have shown that the GNPs distribution in the tumor is bound to be non-uniform, with hypoxic tumor regions receiving lower concentrations. For a nonuniform distribution of the GNPs, 3D conformal radiation therapy (3DCRT) is not able to produce acceptable dose distributions and, therefore, more advanced techniques, such as small field stereotactic body radiation therapy (SBRT) should be the method of choice [28]. In this radiotherapy technique, the typical dose fraction is in the order of 10 Gy and, therefore, it is precisely this absorbed dose value that will be taken as the reference dose to be delivered to the tumor phantom. To do this, for each energy the dose matrices without GNPs present yielded by PENELOPE were renormalized to obtain a minimum phantom dose of 10 Gy, and the normalization factor thus obtained was then applied to the dose matrices obtained when the phantom was irradiated with the GNPs at the 10 mg-Au/g concentration level.

## 3. Results

Fig 1 shows a section of the tumor vasculature model developed with the algorithm described above, with four structures or regions of interest clearly discernible: the tumora tissue, exterior and interior blood vessel walls and blood. In this model, the vessel wall thickness is 2 μm and the vasculature radii at different segments in the modeled volume range from 2 to 20 μm, representing capillaries of different diameters. The total vasculature volume is 3% of the total phantom volume [22].

### a. Dose enhancement when GNPs circulate in the bloodstream

Fig 2 shows cumulative dose-volume histograms (cDVH) in each structure of interest, namely the blood, vessel walls and surrounding tissue when the GNPs are circulating exclusively in the bloodstream and also for the reference case when no GNPs are present during the irradiation. This situation can be considered to occur immediately and up to 2 h after the IV administration of the GNPs. The data were normalized so that the minimum phantom dose was least 10 Gy, a typical fractionation scheme in SBRT. For the 6 MV linear accelerator spectrum, the presence of the GNPs is barely noticeable while for both the lower energy spectra large shifts in the cDVH curves toward regions of higher absorbed dose can be readily seen, with the increase in dose being more pronounced for the 100 keV monoenergetic beam, as the totality of its x-ray fluence has an energy close to the k-edge absorption of 80.7 keV for gold. In order to systematically quantify the effect that the presence of the GNPs has on the resultant absorbed dose distributions, we use the dose-enhancement ratio (DER), that is the ratio of doses with

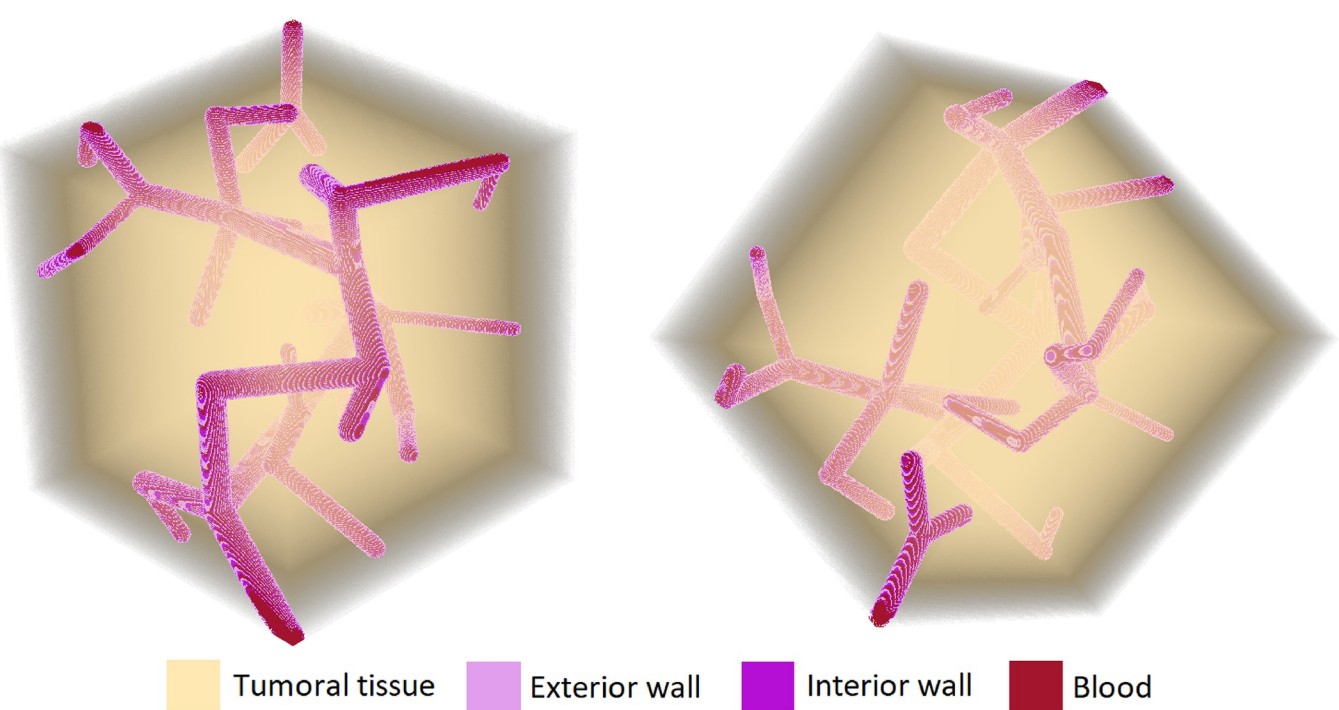

**Fig 1. Tumor vasculature model developed for this work.** The cubic volume has dimensions of 1 mm$^3$. The ratio of vasculature to total volume is 3%.

and without the GNPs present, calculated at three different points in the cDVHs curves, namely at volumes 25%, 50% and 75%.

Fig 3 shows the DERs for the three energy spectra modeled in this work when the GNPs are circulating in the bloodstream for the three structures of interest examined in this work. For the 6 MV flattening-filter free linear accelerator, the average DER, taken over the three sampled points in the DVH curve, is 1.05, 1.04, and 1.04 for the blood itself, the vessel walls and the tumoral tissue respectively. These DER values for the 6 MV beam are consistent with those reported in the literature [14], so even though the removal of the flattening filter yields a softer x-ray spectrum than a conventional linear accelerator [26], the enhancement effect for the megavoltage spectrum is marginal. For the 220 kVp x-ray beam, substantial dose enhancement is observed in the three structures with average DERs of 4.9, 4.5 and 3.4 in the blood itself, the vessel walls and the tumoral tissue respectively. The highest effect is observed when the 100 keV is used to irradiate the tumor volume, with average DERs of 6.9, 5.4 and 4.5. Note that for an average dose fraction of 10Gy under reference conditions without the GNPs present, these DER values would translate into dose enhancements of up to 54 Gy and 45 Gy imparted to the blood vessel walls and the tumor tissue located in the 1 mm$^3$ surrounding the tumor vasculature. This level of absorbed dose to the blood vessel walls in particular would be enough to collapse them [10].

## b. Dose enhancement for GNPs attached to the blood vessel walls

Fig 4 shows the cDVHs for blood, vessel walls and the surrounding tumoral tissue when the GNPs are attached to the vessel walls as well as for the reference case when no GNPs are present, for all three beam energies.

The attachment of the GNPs to the interior vessel walls can be expected to occur during the GNPs circulation in the bloodstream, prior to the extravasation by means of the permeability

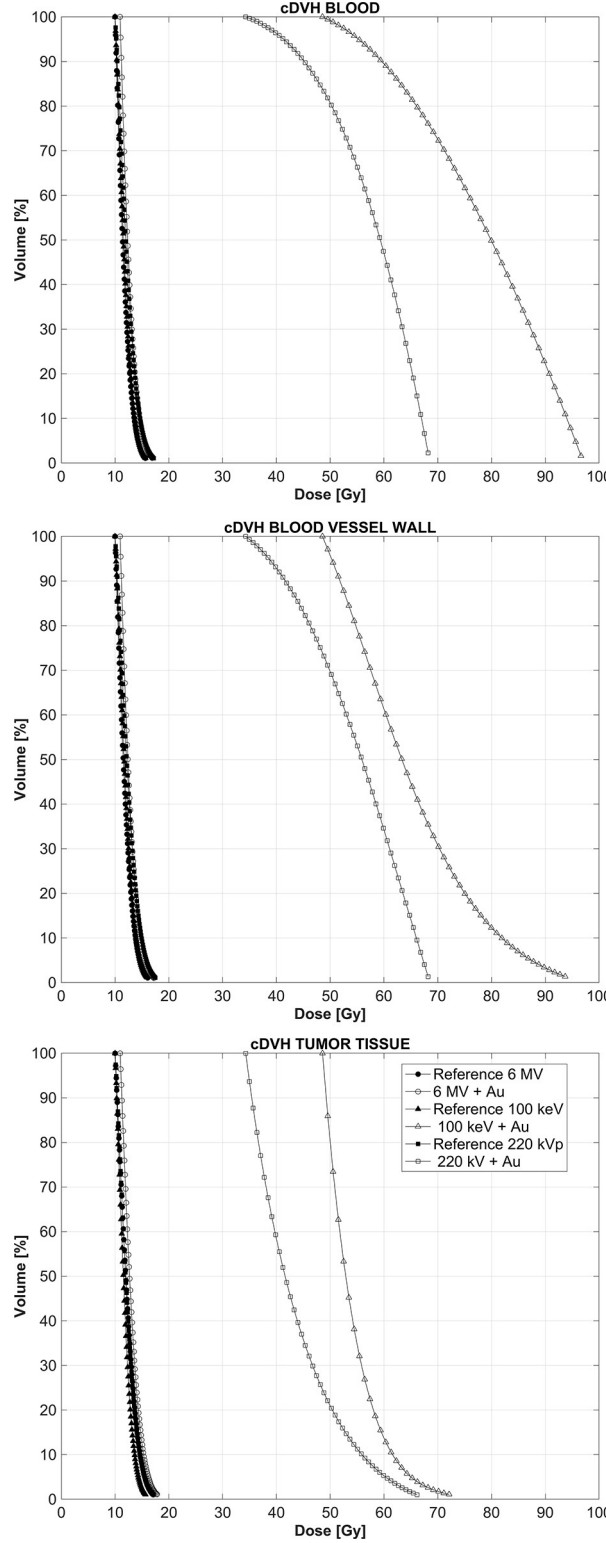

**Fig 2. cDVHs for the three structures of interest and the three energy spectra modeled, when the GNPs are circulating in the bloodstream.** "Reference" refers to the results when there are no GNPs present during the irradiation.

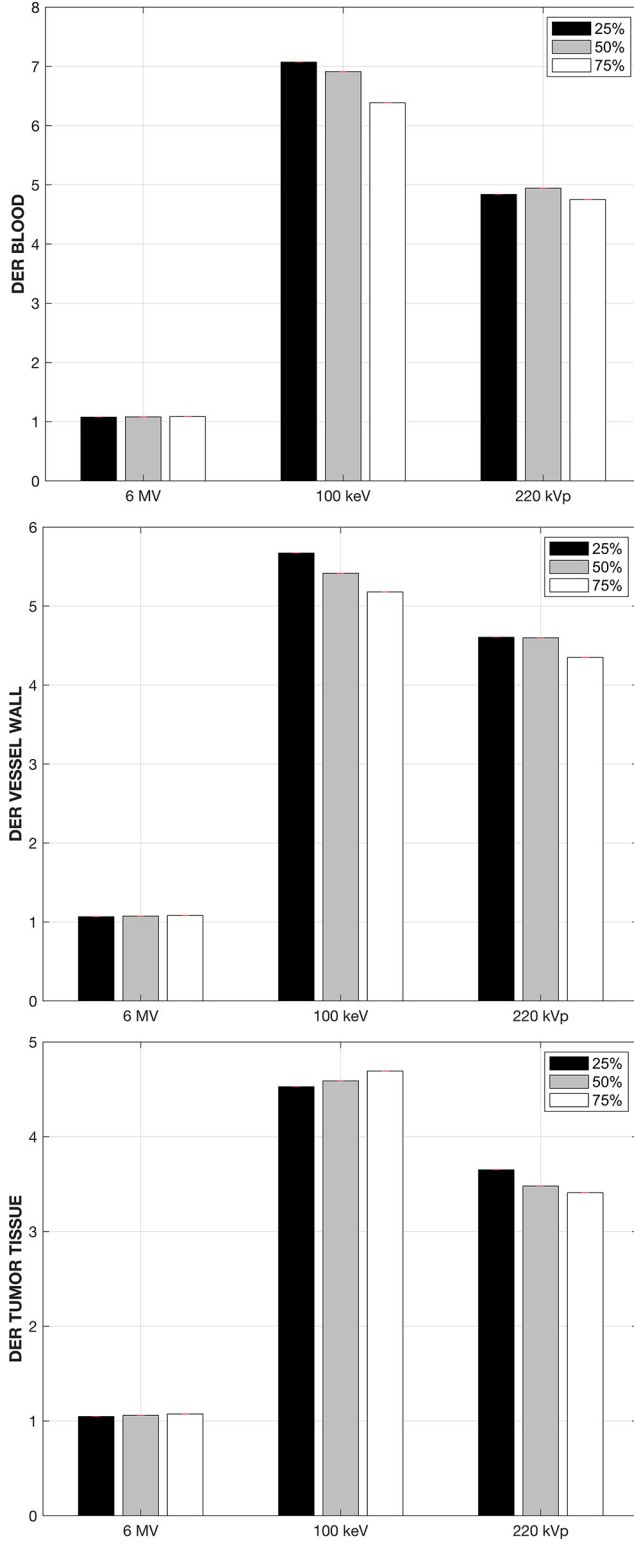

**Fig 3. Dose-enhancement ratio (DER) when the GNPs are circulating in the bloodstream, shown for each energy spectra modeled in this work and for the three structures of interest, namely the blood, vessel walls and the tumoral tissue.**

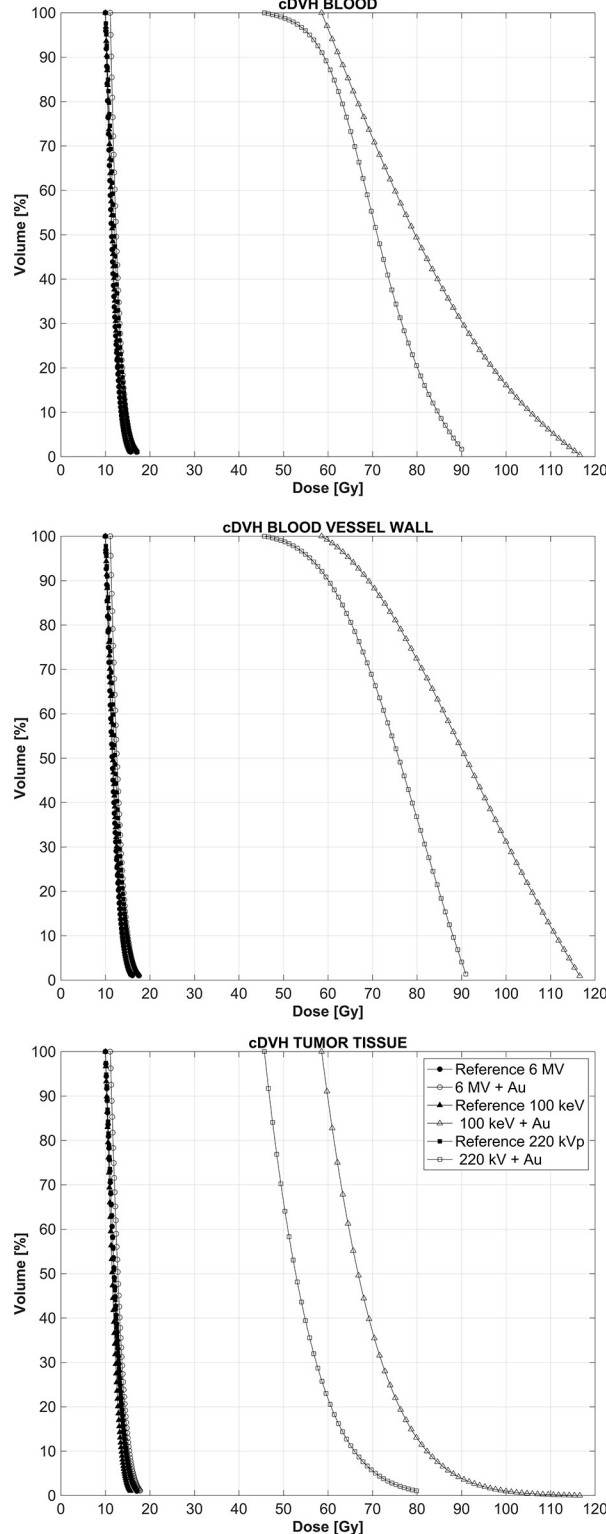

**Fig 4. cDVH for the three regions of interest resulting from a GNPs distribution over the vessel walls only.**

and retention effect. Note that we are considering that the total GNP concentration in the 1mm³ tumor phantom, 10 mg-Au/g, are attached to the vessel walls. As was the case when the GNPs were in the bloodstream, large spatial non-uniformities in the absorbed dose distributions are observed for both kilovoltage beams in both the bloodstream and the vessel walls themselves, and to a lesser degree in the tumoral tissue. Fig 5 shows the DER obtained at three different points in the cDVH curve for the three structures of interest, when the GNPs are attached to the inner vessel walls. For the 6 MV linear accelerator beam the maximum enhancement observed is in the order of 1.04 or less regardless of the structure of interest.

The 100 keV beam again yields the highest DER values of 6.9, 7.9 and 5.8 in the bloodstream, vessel walls and tumoral tissue respectively, followed by the 220 kVp x-ray beam. Note that in both situations when the GNPs are confined to the vasculature the tumoral tissue still experiences a high increase in the absorbed dose for both kilovoltage beams, with DER values of at least 3.5 and 4.5 for the 220 kVp and 100 keV beam respectively.

## c. Dose enhancement when GNPs are uniformly distributed in tissue

Fig 6 shows the cDVH for the three structures of interest when the GNPs are uniformly distributed in the tumoral tissue only, which can be expected to occur at least 2 h after the IV administration of the GNPs [5]. In this case, as the GNPs are uniformly distributed and the vasculature volume represents only 3% of the total irradiated volume, for each energy spectrum there basically is no difference in the dose distributions in the three structures of interest. Importantly, the shift to higher doses for both kilovoltage beams, while present and substantial, is significantly less pronounced when compared to the case when the GNPs were confined to the tumor vasculature, either in the bloodstream or attached to the internal vessel walls.

Fig 7 shows the DERs that results from an irradiation of the tumoral volume with a uniform spatial distribution of the GNPS, further showing that all three regions of interest exhibit essentially the same absorbed dose distribution. The largest average DER of 2.5 is again obtained when the irradiation is carried out with the 100 keV x-ray beam, with the 220 kVp beam yielding an average DER of 1.8, substantially lower when compared to those obtained when the irradiation takes place with the GNPs confined to the tumor vasculature, which yielded DER values in the order of 9. This effect will be discussed below, but it is important to note that the practical consequence of this is that measurement of the gold nanoparticle concentration in a given tumor volume is not enough for an accurate computation of the absorbed dose distribution, as this distribution will depend on the microscopic location of the nanoparticles, be this the tumor vasculature or the tissue surrounding it. Furthermore, given the dynamics of the transport of the GNPs between administration and extravasation into the tumor tissue, there is a time window under which the irradiation must be carried out in order to maximize the dose enhancement effect produced by the presence of the GNPs before they extravasate.

Fig 8 shows a color wash plot of a cross-sectional view of the vascular phantom developed in this work, for the three energy spectra modeled and the three spatial distributions of the GNPs at the analyzed concentration of 10 mg-Au/g. The same cross-sectional view is shown in all panels. Table 1 shows the average absorbed dose taken over the tumor tissue for each energy and spatial distribution of the GNPs in the bloodstream, the blood vessel walls and the tumor tissue itself. Clearly, the more confined the GNPs are in the irradiated volume, the higher the average absorbed dose imparted to the tumor tissue. This is a geometric effect that will be discussed below and that, given the influence that it has on the resultant absorbed dose distribution, it is important to consider when calculating the dose enhancement in a tumor loaded with metallic nanoparticles.

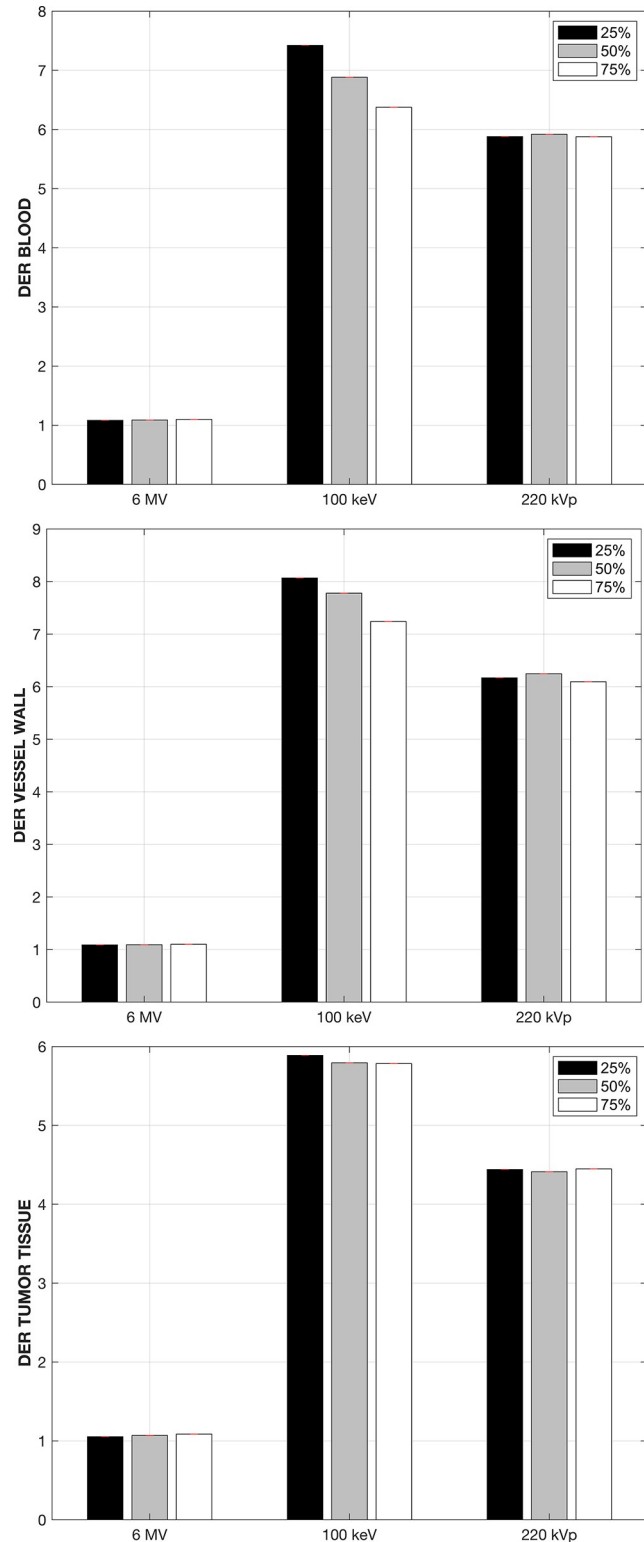

**Fig 5. Dose-enhancement ratio in the three tissues of interest when the GNPs are attached to the blood vessel walls.**

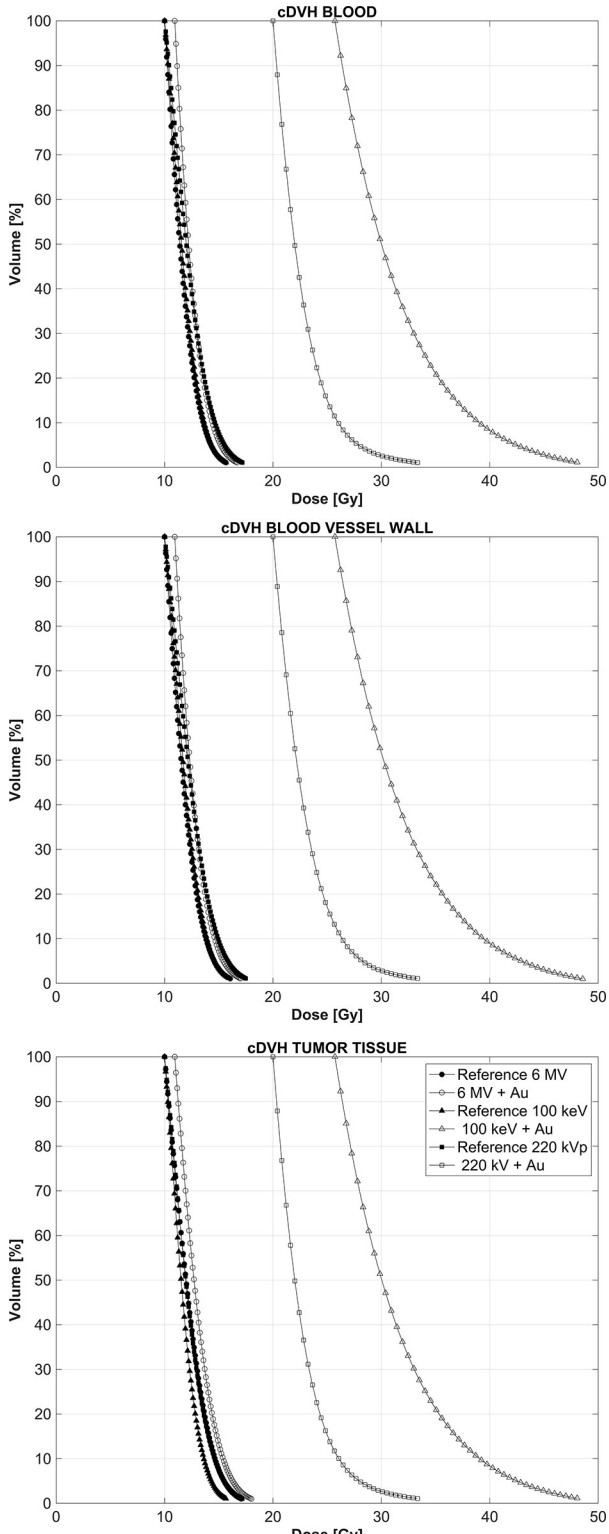

**Fig 6. cDVH for the three tissues of interest when the GNPs are uniformly distributed in the tumoral tissue.**

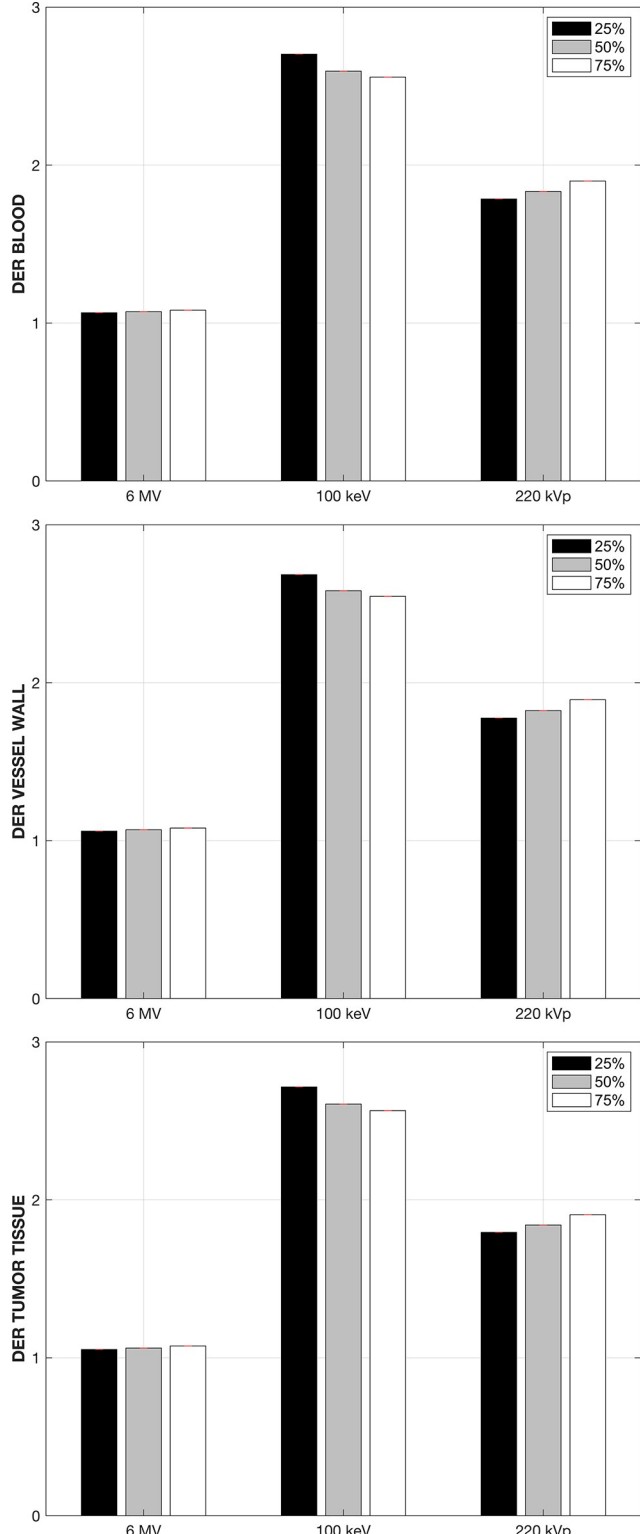

**Fig 7. Dose-enhancement ratio in the three tissues of interest when the GNPs are embedded in the tissue surrounding the tumor vasculature.**

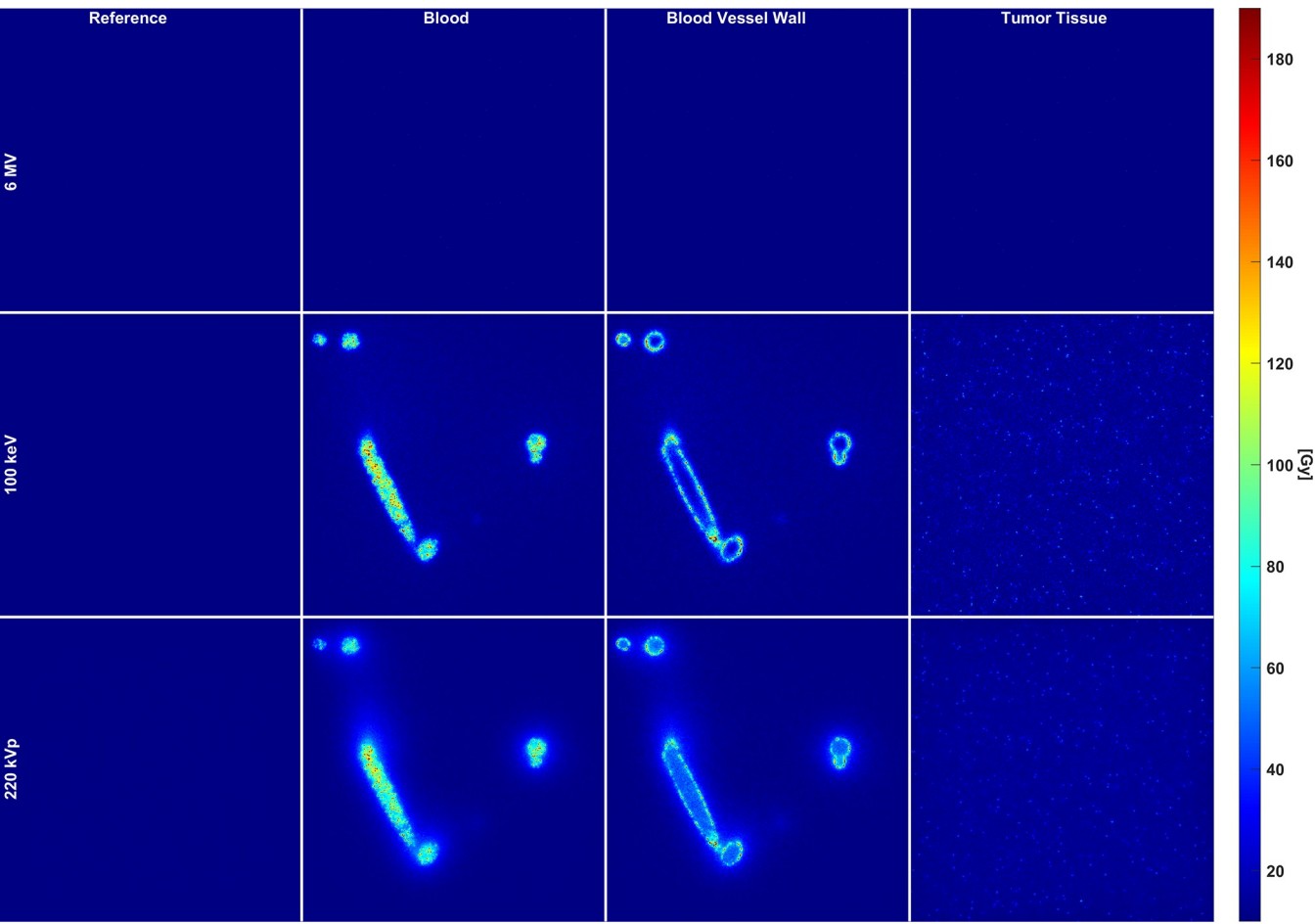

**Fig 8. A cross sectional view of the vascular phantom developed in this work showing the effect that the presence of the GNPs has on the absorbed dose distribution, when they are in the bloodstream (second column), the internal vessel walls (third column) and the tumor tissue (rightmost column).** The first column shows the reference case when no GNPs are present in the vascular phantom.

## 4. Discussion

In practice and depending on the particular time at which the radiation is delivered, the spatial distribution of the GNPs would be a complex mix of the three separate situations examined in this work, with GNPs simultaneously circulating in the bloodstream, attached to the vessel walls and extravasated into the tumoral tissue. But about 2 h after the intravenous administration [5], it can be expected that the GNPs will be located in the tumor tissue only, although it is likely that the exact time for this will depend on the tumor type. It is clear from the results

**Table 1. Average absorbed dose (Gy) in the tumor tissue for each energy modeled in this work.** The minimum dose under reference conditions is 10 Gy.

|  | No GNPs | GNPs | | |
|---|---|---|---|---|
| ENERGY | REFERENCE | BLOOD | VESSEL WALLS | TUMOR TISSUE |
| 6 MV | 13.5 | 14.4 | 14.5 | 14.5 |
| 100 keV | 12.8 | 60.3 | 87.6 | 37.0 |
| 220 kVp | 13.6 | 50.2 | 62.8 | 26.8 |

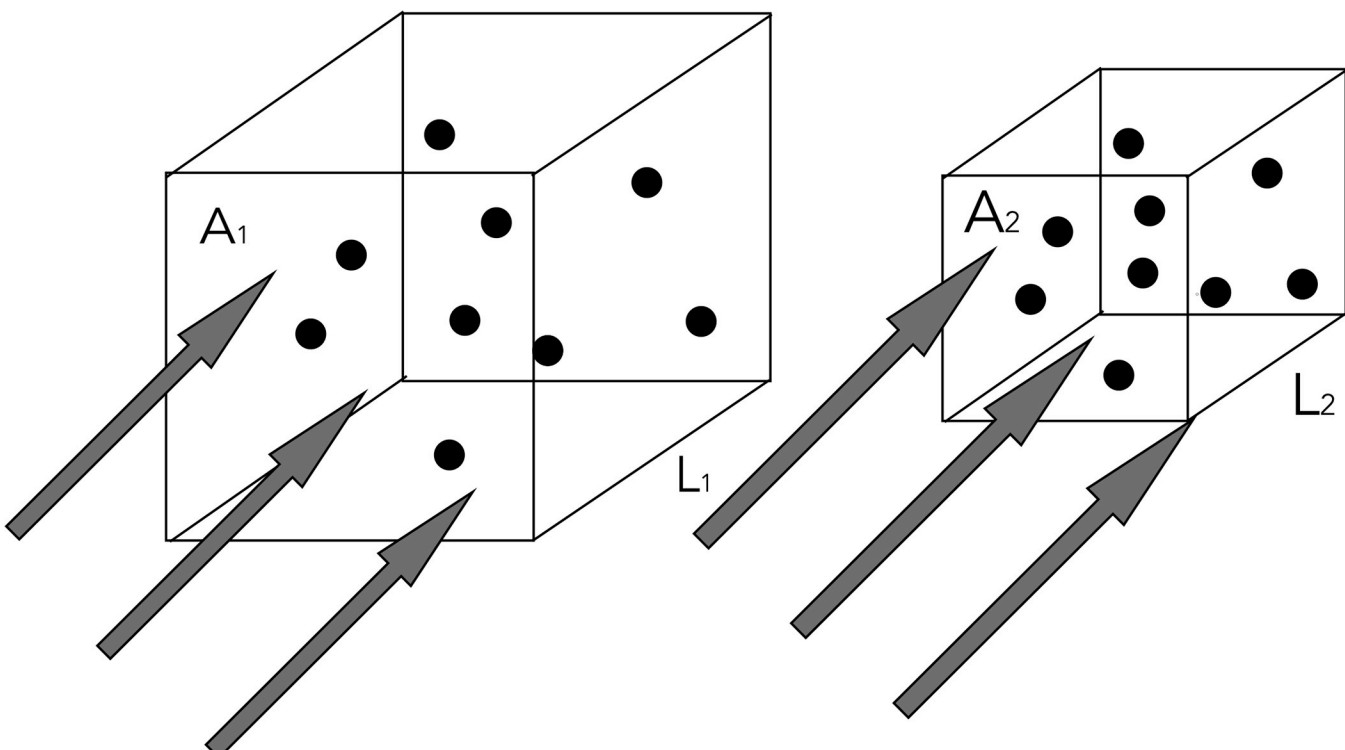

**Fig 9. The same number N of GNPs with the same dimensions irradiated under different conditions of confinement by the same photon fluence.**

above that in order to maximize the damage to the tumor tissue, the radiation must be delivered when the GNPs are confined to the tumor vasculature. If the time between IV administration of the nanoparticles and their extravasation into the tumor tissue is indeed in the order of 2 h, this would be enough to carry out the radiation delivery in order to maximize the tumor tissue damage. The challenge of course is to be able to determine the proportion of GNPs that are inside and outside the tumor vasculature at any given time, as the enhancement in the absorbed dose, and consequently the damage to both the blood vessel walls and the surrounding tumor tissue depends on this proportion. This necessitates the development of mathematical tools that, based on measurable parameters of the GNP transport in blood vessels, provide an accurate estimation of the percentage of the measured GNP concentration in a given tumor that is located in each of these three structures. Regarding the seemingly counterintuitive result that the enhancement in the tumor tissue dose is larger when the GNPs are confined to the tumor vasculature as opposed to in the tumor tissue itself, consider the two situations depicted in Fig 9, which shows the same fluence Φ of x-rays uniformly incident on two volumes with cross sectional areas A and depth L. For the sake of illustration, it is assumed that each volume is occupied by the same number N of nanoparticles. The probability of an interaction between a photon and a GNP is composed of two multiplicative terms: a geometric probability that the GNP is in the rectilinear trajectory of the photon and a radiological probability that, if a GNP is indeed located in the path of the incident x-ray photon, an interaction will take place. For a uniform photon fluence and distribution of the GNPs, the former is given by the ratio of volumes between the total number of GNPs and the irradiated volume, and the latter is given by the well-known exponential attenuation formula. Mathematically, the probability of interaction between the photon fluence and the GNPs in the volumes with cross sectional area A and

volume V, shown in Fig 9, is given by:

$$P_i = \Phi \cdot A_i \cdot \left( N \cdot \frac{V_{GNP}}{V_i} \right) \cdot e^{-\mu D}; \; i = 1, 2 \tag{1}$$

Where $\Phi \cdot A_i$ is the number of x-ray photons available for the interaction process, the term in brackets is the geometric probability and the exponential term, where $\mu$ is the linear attenuation coefficient of gold, is the radiological probability of interaction with a GNP of diameter D and volume $V_{GNP}$. The ratio of probabilities for the volumes $V_1 = A_1 \times L_1$ and $V_2 = A_2 \times L_2$ depicted in Fig 9 is then:

$$\frac{P_2}{P_1} = \frac{L_1}{L_2} \begin{cases} > 1, & L_2 < L_1 \\ < 1, & L_2 > L_1 \end{cases} \tag{2}$$

Therefore, for a uniform photon fluence the confinement of the GNPs into a smaller volume leads to an increase in the probability of interaction and thus to higher DER values, as has been shown above.

In this work, we have addressed the practical implications that arise when the actual distribution of gold nanoparticles within a tumor model that considers the vasculature is taken into account when determining the tumor dose enhancement. It is clear from our results that it is not sufficient to quantify the macroscopic distribution of the nanoparticles via radiological imaging, but additional tools that model the distribution of the nanoparticles at the microscopic level are needed if the dose enhancement is to be fully exploited. Several other aspects need to be taken into consideration as well before this treatment modality can be translated to the clinic. In particular, if the tumor vasculature is compromised, hypoxic regions in the tumor may develop with a higher resistance to x-ray radiation damage [29]. This would of course represent a problem if subsequent treatment fractions were to be delivered, but it would be irrelevant if the total dose is applied in a single fraction. Therefore, proper fractionation schemes need to be determined as well.

## 5. Conclusions

Using a tumor vasculature model with a resolution of 500 nm upon which a given concentration of gold nanoparticles is incorporated either into the bloodstream, attached to the blood vessel walls or uniformly distributed in the tumor tissue, we have shown that upon irradiation with x-rays of three different energy spectra, the largest dose enhancement in the tumor phantom occurs when the nanoparticles are confined to the tumor vasculature. Therefore, it is not enough to measure the concentration of gold nanoparticles in a tumor volume in order to determine the resultant absorbed dose, but also their distribution in the tumor vasculature or tissue needs to be determined and taken into account in order to perform the tumor dosimetry with the accuracy needed to guarantee a successful radiotherapy treatment.

## Author Contributions

**Conceptualization:** H. M. Garnica-Garza.

**Investigation:** C. A. Díaz-Galindo, H. M. Garnica-Garza.

**Methodology:** C. A. Díaz-Galindo.

**Project administration:** H. M. Garnica-Garza.

**Software:** C. A. Díaz-Galindo, H. M. Garnica-Garza.

**Writing – original draft:** H. M. Garnica-Garza.

**Writing – review & editing:** C. A. Díaz-Galindo, H. M. Garnica-Garza.

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
