## [Decision Letter · Decision Letter 0]

30 Apr 2024

PONE-D-24-06743Gold nanoparticle-enhanced radiotherapy: Dependence of the macroscopic dose enhancement on the microscopic localization of the nanoparticles within the tumor vasculaturePLOS ONE

Dear Dr. Garnica-Garza,

Thank you for submitting your manuscript to PLOS ONE. After careful consideration, we feel that it has merit but does not fully meet PLOS ONE’s publication criteria as it currently stands. Therefore, we invite you to submit a revised version of the manuscript that addresses the points raised during the review process.

We look forward to receiving your revised manuscript.

Kind regards,

Tanveer A. Tabish

Academic Editor

PLOS ONE

Journal Requirements:

Reviewers' comments:

Reviewer's Responses to Questions

**Comments to the Author**

1. Is the manuscript technically sound, and do the data support the conclusions?

Reviewer #1: Partly

Reviewer #2: Yes

2. Has the statistical analysis been performed appropriately and rigorously? 

Reviewer #1: N/A

Reviewer #2: Yes

3. Have the authors made all data underlying the findings in their manuscript fully available?

Reviewer #1: Yes

Reviewer #2: Yes

4. Is the manuscript presented in an intelligible fashion and written in standard English?

Reviewer #1: Yes

Reviewer #2: Yes

5. Review Comments to the Author

Reviewer #1: 1. What is the rationale for selecting vasculature model... It should be compared with other equivalent models and related work. Further this is simulation work and considered volume is too small and relatively unrealistic compared with the real tumor.

2. The three different regions are considered for distribution of Gold NPs. How the results will change with considering a gradient in distribution?

3. What is the maximum limit of dose and corresponding damages?

4. The impact of radiation dose with other NPs may be compared and selection of Gold NPs should be justified...

Reviewer #2: This is a nicely written paper. The research subject is of substantial interest to readers of PLOS One showing the tumor vasculature model to determine the exact concentration of gold nanoparticles as a dose with the accuracy needed to confirm a successful radiotherapy treatment.

I find the paper to be thorough and the conclusions well grounded. Overall, I commend the authors on the high quality of their research and this manuscript.

6. PLOS authors have the option to publish the peer review history of their article (what does this mean?). If published, this will include your full peer review and any attached files.

Reviewer #1: No

Reviewer #2: **Yes**

---

## [Author Response · Author response to Decision Letter 0]

14 May 2024

REVIEWER 1

We thank the anonymous reviewer for his comments and observations. We hope that the answers we provide below address each of his/her queries. 

1. What is the rationale for selecting vasculature model? It should be compared with other equivalent models and related work. Further this is simulation work and considered volume is too small and relatively unrealistic compared with the real tumor.

The vasculature model was developed specifically to carry out the research reported in this work, as mentioned in section 2.a in the manuscript. Strictly speaking, the model was not “selected” per se, as a random algorithm was used to develop it, so we did not know a priori what the final model would look like. But the parameters fed into the random algorithm model were such that a tumor volume with a vasculature displaying the main features that have previously been reported in the literature would result. As for the size of the volume modeled, it must be kept in mind that using a resolution of 500nm for dividing a cube of 1 mm side implies that a total of 8 x 109 voxels would be used in the simulations. Taking into account that PENELOPE (the Monte Carlo radiation transport code used in our work) uses double precision variables, 64 GB of RAM are needed just to store the tumor model. Similar amounts of computer memory are also needed to store the dose matrix which was calculated in each tumor voxel. This would pose a serious challenge, in terms of computer resources, if larger tumor models were to be considered. Furthermore, the more voxels are added to the tumor model, the longer the computer time it takes to complete the calculation. As they were run, our simulations took between 10 and 15 days of computer time, sometimes even longer, particularly for the 6 MV beam. However, we would like to emphasize that we do not believe the proposed tumor model limits in any way the conclusions reached in our work, as the following steps were taken in order to ensure that the results extracted from our model were meaningful:

a. The random algorithm used to grow the tumor vasculature was setup in such a way ratio of vasculature to total tumor volume is in the order of 3%, as has been reported to be the case in real tumors (see reference 22 in the manuscript). Also, the radial dimensions of the vasculature were set in accordance to published data. This ensures that, regardless of its size, a realistic tumor volume is being modeled from a physiological point of view. 

b. Materials such as soft tissue and blood were used in the tumor model, with their composition taken from ICRU report 44. 

c. The tumor model is embedded in a spherical tissue sample with dimensions large enough to provide charged particle equilibrium throughout the tumor for all the beam energies modeled in our work. This means that for a given radiation field, the resultant absorbed dose distribution is not affected by the particular geometric details of such a region of interest, but rather by its composition. 

d. The gold nanoparticles were explicitly incorporated into the tumor model, and their clustering, an effect that has important consequences on the resultant absorbed dose distribution (see for example references 16 and 17 in our manuscript) was also considered. 

e. The 1 mm3 volume is the typical size of the minimum element (voxel) on which the absorbed dose is computed in radiation therapy, where they used CT or MRI images to model the patient geometry and perform the absorbed dose calculations. 

Regarding the comparison with equivalent models and related work: to the best of our knowledge, this is the first report that considers a realistic vasculature microscopic model to compute the absorbed dose in a macroscopic volume. As mentioned in the manuscript, previous work on this subject involved the use of a simplified straight, cylindrical blood vessel upon which a given number of GNPs were placed, either in the cylinder´s surface or in close proximity to it (see reference 11 in the manuscript) and they focused their calculations on the blood vessel wall. Therefore, there is no data available in the literature that we can use as a reference against which to compare our results. This was in fact one of the reasons why this research project was started. 

Finally, as the reviewer notes, this is indeed a simulation work. But we can’t think of any other way in which absorbed dose measurements at the vasculature level and with a resolution of 500 nm could be carried out. No instrument offers this resolution and no instrument is capable of measuring radiation doses in tissue samples without compromising such samples. Monte Carlo simulation of the radiation transport is particularly well suited for this task and, in our opinion, is the only tool available to attack this kind of problems. 

In order to emphasize the suitability of the model used for the research reported in this manuscript, the following paragraph has been added to section 2.a of the revised manuscript: “While the tumor vasculature model is small in size, a necessity dictated by its high resolution, the fact that it correctly reproduces the ratio of vasculature-to-tissue volume, that it is composed of biological materials, that it is embedded in a soft tissue sphere in order to achieve electronic equilibrium inside of it, and that it is, nevertheless, large enough to allow us to calculate absorbed dose at the dimensions typically used in radiotherapy treatment planning, makes this model suitable for the purpose of determining the effect that the presence of GNPs have on the absorbed dose imparted not only to the tumor tissue but also to the blood vessel walls.” 

2. The three different regions are considered for distribution of Gold NPs. How the results will change with considering a gradient in distribution?

The only possible gradient distribution could present itself in the tissue surrounding the tumor vasculature, as both the blood vessel walls and the bloodstream are too narrow to allow for a meaningful spatial gradient to be obtained. In the tumor tissue, the GNPs accumulate as they extravasate and diffuse from the tumor vasculature, which follows a random path throughout the tumor volume. For a rectilinear blood vessel, a gradient distribution of the GNPs would be possible, with higher concentrations near the vessel wall and lower concentration away from it. But for a random web of blood vessels, the gradient distribution does not make sense, because the vasculature path is convoluted and does not follow a spatial pattern that may allow for a gradient to form. As soon as the concentration decreases in a given direction, GNPs extravasating from another segment of the vasculature would prevent the gradient to form. Because of this, we decided that for the modeling of the GNP distribution in the tumor tissue, the best way to proceed was to uniformly (but randomly) seed the GNPs in the tissue. 

In order to make the reader aware of the rationale for uniformly distributing the GNPs in the tumoral tissue, we have added the following paragraph in section 2.b of the revised manuscript: “While it is likely that once GNPs extravasate from the vasculature and diffuse in the surrounding tissue a concentration gradient would form, this would only be the case if the blood vessels are rectilinear and fairly parallel to each other. For the inherently random tumor vasculature, we decided that the most representative approach would be to consider a uniform spatial distribution of the GNPs in the tumor tissue.” 

3. What is the maximum limit of dose and corresponding damages?

The maximum dose limit depends on the organ or structure being irradiated. As mentioned in the manuscript, for the particular case of blood vessel walls, it has been shown that an absorbed dose of 10 Gy imparted in a single fraction can result in severe vascular damage (see reference 10 in the manuscript, for example). In this work we have shown that when GNPs are present in the vicinity of the blood vessel walls, depending on the energy of the x-ray beam, absorbed doses of up to 86 Gy can be delivered under irradiation conditions that, without the GNPs present, would result in doses in the order of 10 Gy. This suggests that it is possible to drastically reduced the amount of radiation used in the radiotherapy treatment while still imparting therapeutically relevant absorbed doses to the vessel walls and the surrounding tumoral tissue when GNPs are present. However, as was also shown in our work, it is important to determine the spatial concentration of the GNPs at the microscopic level, i.e. are they attached to the blood vessel walls, circulating in the bloodstream or deposited in the surrounding tissue, in order to be able to quantify the resultant absorbed dose distribution at the macroscopic level. By macroscopic level mean the dimensions at which typically the absorbed dose is computed in radiotherapy, which is in the order of 1 mm3, as was modeled in this work. 

4. The impact of radiation dose with other NPs may be compared and selection of Gold NPs should be justified.

Gold nanoparticles were chosen in this work as it is already known that gold has the highest interaction probability compared to other high-atomic nanoparticles such as gadolinium- or iodine-based, which are undergoing several clinical trials to evaluate their effectiveness as radiation enhancers. This means that gold produces the highest dose enhancement in the surrounding tissue. Gold also has the advantage that it is chemically inert and therefore does not produce adverse reactions in humans. We do not believe that there is a need to compare between different materials, as this has already been done, see for example “A Monte Carlo comparison of three different media for contrast enhanced radiotherapy of the prostate” Technol Cancer Res Treat 2010 9 271-8, where gold, gadolinium and iodine are compared in terms of the dose enhancement they produce when irradiated with kilovoltage x-rays. Gold yields the higher dose enhancement per incident x-ray.

REVIEWER 2

This is a nicely written paper. The research subject is of substantial interest to readers of PLOS One showing the tumor vasculature model to determine the exact concentration of gold nanoparticles as a dose with the accuracy needed to confirm a successful radiotherapy treatment.

I find the paper to be thorough and the conclusions well grounded. Overall, I commend the authors on the high quality of their research and this manuscript.

We thank the reviewer for taking the time to assess our work and for their kind comments regarding our research.

---

## [Editor Report · Decision Letter 1]

16 May 2024

Gold nanoparticle-enhanced radiotherapy: Dependence of the macroscopic dose enhancement on the microscopic localization of the nanoparticles within the tumor vasculature

PONE-D-24-06743R1

Dear Dr. Garnica-Garza,

We’re pleased to inform you that your manuscript has been judged scientifically suitable for publication and will be formally accepted for publication once it meets all outstanding technical requirements.

Kind regards,

Tanveer A. Tabish

Academic Editor

PLOS ONE
---

## [Editor Report · Acceptance letter]

24 Jun 2024

PONE-D-24-06743R1 

PLOS ONE

Dear Dr. Garnica-Garza, 

I'm pleased to inform you that your manuscript has been deemed suitable for publication in PLOS ONE. Congratulations! Your manuscript is now being handed over to our production team.

Kind regards, 

on behalf of

Dr. Tanveer A. Tabish 

Academic Editor

PLOS ONE